# *GmUFO1* Regulates Floral Organ Number and Shape in Soybean

**DOI:** 10.3390/ijms24119662

**Published:** 2023-06-02

**Authors:** Huimin Yu, Yaohua Zhang, Junling Fang, Xinjing Yang, Zhirui Zhang, Fawei Wang, Tao Wu, Muhammad Hafeez Ullah Khan, Javaid Akhter Bhat, Yu Jiang, Yi Wang, Xianzhong Feng

**Affiliations:** 1College of Life Sciences, Jilin Agricultural University, Changchun 130118, China; 2Key Laboratory of Soybean Molecular Design Breeding, Northeast Institute of Geography and Agroecology, Chinese Academy of Sciences, Changchun 130102, China; 3University of Chinese Academy of Sciences, Beijing 100049, China; 4Zhejiang Lab, Hangzhou 311121, China

**Keywords:** *GmUFOs*, knockout, floral organ number, floral organ shape, soybean

## Abstract

The *UNUSUAL FLORAL ORGANS* (*UFO*) gene is an essential regulatory factor of class B genes and plays a vital role in the process of inflorescence primordial and flower primordial development. The role of *UFO* genes in soybean was investigated to better understand the development of floral organs through gene cloning, expression analysis, and gene knockout. There are two copies of *UFO* genes in soybean and in situ hybridization, which have demonstrated similar expression patterns of the *GmUFO1* and *GmUFO2* genes in the flower primordium. The phenotypic observation of *GmUFO1* knockout mutant lines (*Gmufo1*) showed an obvious alteration in the floral organ number and shape and mosaic organ formation. By contrast, *GmUFO2* knockout mutant lines (*Gmufo2*) showed no obvious difference in the floral organs. However, the *GmUFO1* and *GmUFO2* double knockout lines (*Gmufo1ufo2*) showed more mosaic organs than the *Gmufo1* lines, in addition to the alteration in the organ number and shape. Gene expression analysis also showed differences in the expression of major ABC function genes in the knockout lines. Based on the phenotypic and expression analysis, our results suggest the major role of *GmUFO1* in the regulation of flower organ formation in soybeans and that *GmUFO2* does not have any direct effect but might have an interaction role with *GmUFO1* in the regulation of flower development. In conclusion, the present study identified *UFO* genes in soybean and improved our understanding of floral development, which could be useful for flower designs in hybrid soybean breeding.

## 1. Introduction

Cell division control is necessary for the proper arrangement of flower primordia and is regulated by the plant hormones auxin and cytokinin. The vast majority of angiosperm flowers have four-wheeled floral organs, which consist of sepals, petals, stamens, and the carpel, in that order, moving from the outside to the interior of the flower. The well-known “ABC model” is the functional model of floral organ attribute genes in *Arabidopsis thaliana* [1,2]. This model correctly describes the molecular process of flower organ primordium attributes and has been the subject of a great deal of research. The A functional genes *AP1* (*APETALA1*) and *AP2*, B functional genes *AP3* and *PI* (*PISTILLATA*), C functional gene *AG* (*AGAMOUS*), and E functional gene *SEP* (*SEPALLATA*) are the most typical genes studied in model plants [3]. This well-known “ABC model” is a scientific model of the process through which a pattern of gene expression can be produced by flowering plants in their meristems, which, in turn, leads to the appearance of an organ that is oriented toward sexual reproduction [4]. This model correctly describes the molecular process of flower organ primordium attributes and has been the subject of a great deal of research.

The *UFO* gene is an F-box gene that encodes E3 ubiquitin ligase that was first discovered in *Arabidopsis thaliana*, where mutant flowers lack typical petals and stamens [1]. In *Arabidopsis thaliana*, the *UFO* gene co-activates the expression of the downstream *AP3* gene with the *AP1* gene as a co-activator of the *LFY* gene, thus preserving petal characteristics and petal development. *LFY* controls the temporal specificity of *AP3* expression, while *UFO* and *AP1* control the spatial pattern [5,6]. UFO has been reported to interact with the SCF E3 ubiquitin ligase subunits, ASK1 and CUL1, as well as subunits of the COP9 signalosome, which suggests the possible role of UFO in the ubiquitination of proteins that are involved in flower development [7,8]. The interaction of the UFO and ASK1 proteins may enhance the degradation of the downstream B functional gene *AP3* and the *PI* inhibitor by the *UFO* gene, resulting in the activation and production of the B functional gene [5,9,10]. Strong loss-of-function mutations in the *UFO* gene can cause flower abnormalities in all whorls. In sepals and petals, homeotic changes, organ number, and phyllotaxis are the most severe [1,8]. Sepals replace petals, and carpels replace stamens; however, mosaic organs are also prevalent in both whorls of the *ufo* mutant. *UFO*, similar to its *Antirrhinum* homolog *FIM*, can be expressed in a complicated temporal and geographic pattern throughout floral development [11,12]. In floral meristems, transcription is inhibited in stage 1 and triggered in stage 2 in the central area. By stage 3, the core meristem loses expression but extends laterally in the cone form. By stage 5, expression is confined to the petal primordia and remains there throughout floral organ development. *AP3* and *PI* patterns are formed during stage 3 [13,14] before petal/stamen organ activation during stage 5, supporting UFO’s significance in the B class function [15].

Soybean (Glycine max) is a globally important crop that is a rich source of edible protein and oil [16]. The reproductive period of soybean (starting from flower initiation until maturity) is essential for higher yield and quality [17]. Identifying and understanding the molecular mechanism of the genes involved in the regulation of the reproductive period and flower development in soybeans are key targets for yield and quality improvement. In the present study, we report that *GmUFOs* play a crucial role in regulating flower development, especially in the quantity and characteristics of petals in connection with the development of the stamen. The *GmUFOs* mutation can cause floral organ morphological defects and alter the expression of ABC flower development genes.

## 2. Results

### 2.1. Identification of UFO Genes

In the present study, the *AtUFO* gene (*AT1G30950.1*) of *Arabidopsis thaliana* was used as a query to search its orthologs in soybean. Two orthologues, *Glyma.05G134000* and *Glyma.08G088700*, were identified and named *GmUFO1* and *GmUFO2*, respectively. For phylogenetic analysis, the *GmUFO1* and *GmUFO2*, as well as another nine *UFO* genes from eight species, were used to construct the phylogenetic tree. The phylogenetic alignment showed that *GmUFO1* and *GmUFO2* were more closely related to the *UFOs* of leguminous plants (Figure 1A). Amino acid sequence alignment and protein 3D model analysis showed that the UFO protein domain was relatively conserved in these species (except cotton), and both GmUFO proteins in the soybean had a conserved F-box (Figure 1B). The protein physical property analysis showed that the amino acid length of UFOs in these species was between 357 and 444, and the polypeptide isoelectric points were almost constant (Appendix A).

### 2.2. Spatial and Temporal Expression Profiles of the GmUFO1 Gene in Soybean Flower Development

RNA in situ hybridization was used to detect *GmUFO1* and *GmUFO2* expression in the soybean flower primordium at different development stages (Figure 2A,B and Appendix A). In stage 1, the flower meristem formation stage, *GmUFO1* was expressed in the central area of the whole flower meristem. In stage 2, which represents the sepal primordium formation stage, *GmUFO1* was expressed between the sepal primordium and the carpel primordium, and this signal was also detected near the central dome of the sepal primordium. In stage 3, the petal primordium formation stage, *UFO1* was mainly expressed in the petal primordium, but a few signals were detected in the sepal primordium. Stage 4 represents the stamen and carpal primordium formation stage; in this stage, *UFO1* was expressed in the petal primordia and was almost absent in the elongating sepal primordia. In stage 5, when the formation of all the flower organs occurred, *UFO1* was expressed at the boundary region of the petal between the sepal and stamen. Furthermore, *GmUFO2* showed a similar expression pattern to *GmUFO1* (Appendix A).

The subcellular localization of GmUFO1 was studied, and the GFP fluorescence of the GFP-GmUFO1 fusion protein was located in the nucleus of *Arabidopsis* protoplasts, which was co-localized with AtAHL22 and confirmed as a nuclear marker gene [18] (Figure 2C).

### 2.3. CRISPR/Cas9 System Generated Target Mutations for GmUFO Genes

To obtain the *GmUFO* mutants, the CRISPR/Cas9-mediated genome editing of the *GmUFO* genes was performed, and the target site for both *GmUFO1* and *GmUFO2* was located ahead of the F-box domain (Figure 3A,B). Four independent T0 generation transgenic lines were obtained, and subsequently, 40 T1 generation transgenic lines were generated from the four T0 plants (Appendix A). In the T1 generation, *GmUFOs* were sequenced, and, based on the target sites sequencing data, ten *Gmufo1* single mutants, six *Gmufo2* single mutants, and three *Gmufo1ufo2* double mutants were identified (Figure 3C and Appendix A). These mutants were used for further investigation in this study.

### 2.4. Knockout of the GmUFO1 Gene Caused Abnormal Flower Organs

All 10 mutant plants obtained in *Gmufo1* of the T1 knockout lines showed heterozygosity at the *GmUFO1* target site, whereas no mutation was found at the *GmUFO2* gene, and three of the mutants from different T0 generation lines were selected for further analysis (Figure 4). The *Gmufo1* seedlings showed no obvious differences compared to the wild type (Figure 3D), whereas the floral organ was different from the wild type (Figure 4A). Compared to the wild type, *Gmufo1* showed a floral organ number and shape alterations in the sepals, petals, and stamens, as well as a mosaic flower organ formation with green sepal-like petals (Figure 4A,B).

Furthermore, five ABC function genes, namely *GmAP1a*, *GmAP2*, *GmAP3*, *GmPI04G*, and *GmAG* were selected, and their expression was detected using qPCR in the three *Gmufo1* lines and the wild type. The expression levels of *GmAP1a*, *GmAP3*, *GmPI04G*, and *GmAG* decreased in all three lines, while the expression levels of *GmAP2* increased (Figure 4C). The expression of *GmUFO1* and *GmUFO2* gene transcripts in *Gmufo1* were detected, and it was found that they did not differ from the wild type (Figure 4D).

### 2.5. Knockout of the GmUFO2 Gene Had No Effect on Flower Organs

Six positive plants obtained in *Gmufo2* of the T1 knockout lines showed heterozygosity at the *GmUFO2* target site, whereas no mutants were possessed at the *GmUFO1* gene, and three of them from different T0 generation lines were selected for further analysis (Figure 5). *Gmufo2* seedlings had no obvious differences compared with the wild type (Figure 3E) and did not show differences in their floral organ shape or number (Figure 5A). Further statistics also showed that the number of *Gmufo2* organs to the wild type did not change (Figure 5A,B). The five ABC functional genes (i.e., *GmAP1a*, *GmAP2*, *GmAP3*, *GmPI04G*, and *GmAG*) in the three *Gmufo2* lines were detected using qPCR. The *GmPI04G* expression decreased, while the *GmAP2* expression increased in all three *Gmufo2* lines (Figure 5C). The expression of *GmUFO1* and *GmUFO2* gene transcripts in *Gmufo2* were detected and it was found that they did not differ from the wild type (Figure 5D).

### 2.6. Double Knockout of GmUFO1 and GmUFO2 Had More Serious Defects in Organ Number and Shape

Among the T1 knockout lines, three lines from different T0 generation lines were double mutated and selected for further analysis. In the case of the *GmUFO1* gene, line *Gmufo1ufo2-4* had a 4-bp deletion at the target site, *Gmufo1ufo2-17* had a 1-bp deletion at the target site, and *Gmufo1ufo2-18* showed a 1-bp deletion after the target site. However, in the case of the *GmUFO2* gene, all three selected mutant plants were heterozygotes at the target site (Figure 6). The vegetative appearance showed no obvious difference in the seedlings of double mutants compared with the wild type (Figure 3F), whereas there were more mosaic organs formed with sepal-like petals and petal-like stamens, and the alteration in the organ number and shape of the double mutant was relative to the *Gmufo1* single mutant (Figure 6A, Appendix A).

The three selected double mutant lines showed obvious variations in the number of sepals, petals, and stamens (Figure 6A,B). The expression of the five ABC functional genes (*GmAP1a*, *GmAP2*, *GmAP3*, *GmPI04G*, and *GmAG*) in the three lines was detected using qPCR. The expression levels of all the ABC genes, *GmAP1a*, *GmAP2*, *GmAP3*, *GmPI04G*, and *GmAG*, decreased in all three *Gmufo1ufo2* plants, and this decrease was greater than that in *Gmufo1* (Figure 6C). The expression of *GmUFO1* and *GmUFO2* gene transcripts in *Gmufo1ufo2* was detected, and it was found that they did not differ from the wild type (Figure 6D).

## 3. Discussion

In previous studies, the *ufo* mutant of different plant species has been observed to show the abnormal development of flower organs. However, there was a difference among the *ufo* mutants of various plants in regard to abnormal flower organ development, which may have been due to some differences in the underlying molecular mechanism originating from plant evolution. For example, the single mutant of *UFO* in *Arabidopsis thaliana* (*Atufo*) showed a slight reduction in inflorescences at co-florescence, but the meristem of all primary and co-flowering buds in *Atufo* stopped growing after the appearance of structures that were fused with sepal-carpus organs. The attribute changes in the *Atufo* flower organs were mainly concentrated in the second and third-round organs, and these quantitative changes were reflected in all-around organs. There were always deletions in the second round of early primordium development [19]. In the model legume *Lotus japonicus*, an orthologue of the *Arabidopsis UFO* gene mutation resulted in the *pfo* mutant [20]. In the *pfo* mutant, the flower primordium was unable to form normal flower organs, which resulted in some sepal-like structures with filamentous structures, and the petals failed to develop [20]. In our study, the changes in flower organs in the soybean *ufo* mutants were mainly concentrated in the second whorl organ, and an increase in sepals and a decrease in stamens were also found in the first and third whorl organs. Even the most severe mutant phenotype of *Gmufo* did not show the complete replacement of the second whorl organ with sepals, and sepal-like petals mostly appeared on the ventral part. Previous studies have documented that by controlling the interaction between UFO and LFY, the *UFO* gene could regulate the downstream targets of *AP3* and *PI*, thereby affecting the flower phenotype [8]. In this context, the interaction among the UFO, ABC, LFY, WUS, and CAL genes was explored previously in different studies at the petal formation stage. For example, LFY interacted with UFO, CAL, and AP1, and UFO was expressed downstream of AP1 [21,22,23]. The interaction between LFY and UFO, as well as between LFY and AP1, was mutual and activated each other [22,23]. LFY directly activated *AG*, and *AG* was also activated by WUS, but AG acted as a WUS antagonist [24,25,26]. In addition, AG directly activated *AP3* and *PI*. Moreover, AP2 was an AG antagonist, and AP2 activated *AP3* and *PI* directly [27]. UFO also directly activated *AP3* and *PI* [28]. Based on the above interactions of UFO with ABC and other genes, it was evident that the loss-of-function mutation of the *UFO* gene affected the expression of ABC genes, both directly and indirectly. For example, it indirectly reduced the expression of downstream *AP3* and *PI* by affecting *LFY* expression, which, in turn, affected *AG* and *AP1* expressions [22,29] and directly decreased the expression of downstream *AP3* and *PI* [28]. Our qPCR analysis is consistent with previous findings. It also demonstrated that the knockout of *GmUFO* in *Gmufo1* and *Gmufo1ufo2* soybean mutants reduced the expression of ABC genes, *GmAP1a*, *GmAP2*, *GmAP3*, *GmPI04G*, and *GmAG* (Figure 4C, Figure 5C and Figure 6C). Hence, *GmUFO* could be regarded as a key regulator of ABC function genes in soybean. The ABC model of the flower organ development worked as follows: A alone controlled sepal development, A and B together regulated petal development, B and C together controlled the stamen identity and C regulated carpel development alone. In our study, the *Gmufo1ufo2* and *Gmufo1* mutants showed the abnormal development of the sepals, petals, and stamens (increased sepals, decreased stamens, and the petal looked similar to sepals), but carpel development was normal. Therefore, in agreement with the ABC model, our results also showed that the knockout of *GmUFO* genes reduced the expression of ABC genes, viz., A (*GmAP1a*, *GmAP2*), B (*GmAP3*, *GmPI04G*), and C (*GmAG*) in the *ufo1ufo2* and *ufo1* mutants, which, in turn, led to the abnormal development of sepals, petals, and stamens. Although *GmAG* expression was reduced in the *ufo1ufo2* and *ufo1* mutants, carpel development was still normal in the mutant plants. This suggests that there might be some other interaction factors working together with *GmAG* in the regulation of carpel development in the soybean, which needed to be further explored. The *ufo2* mutant did not show any difference in the flower development compared with the wild type, where only the expression levels of *GmAP2* and *GmPI04G* were decreased in the *ufo2* mutant, and the expression of the remaining ABC genes did not differ relative to the wild type. In addition, our study documented that the effect of the *ufo1ufo2* mutant on flower development was more severe than that of *ufo1*. In addition, although the *ufo* mutants in soybean showed obvious effects on the flower development, this may not have significantly influenced the yield when comparing the seed’s weight per plant of the mutants and wild type growing in the growth chamber under short-day conditions (Appendix A). These findings suggest the major role of *GmUFO1* in the regulation of flower organ formation in soybeans. *GmUFO2* did not have any direct effect on flower development, but it could have an interactive role with *GmUFO1* in the regulation of flower development.

In conclusion, our work demonstrated that *GmUFO* genes regulated the development of flower organs, viz., soybean petals, stamens, and sepals, in the soybean by controlling the expression of ABC genes. These findings should be the basis for further investigation of the in-depth mechanism involved in the *GmUFO*-mediated regulation of flower development in the soybean. This, in turn, could have a significant effect on soybean hybrid breeding, leading to the development of high-yielding cultivars.

## 4. Materials and Methods

### 4.1. Plant Materials and Growth Conditions

The soybean cultival Williams 82, which was genome sequenced as the reference genome and deposited in GenBank under the accession ACUP00000000 [30], was obtained from the Chinese Academy of Agricultural Sciences (Beijing, China) and used for transformation. The transgenic lines and wild-type Williams 82 were planted in a growth chamber under short-day conditions at 25 °C for 12 h of light and 20 °C for 12 h of dark.

### 4.2. GmUFO Identification and Cloning

The *Arabidopsis thaliana UFO* gene (*AT1G30950.1*) sequence was downloaded from The Arabidopsis Information Resource (TAIR) database (https://www.arabidopsis.org/, accessed on 12 December 2020), and this gene was used as a query to identify putative orthologs of *UFO* genes in the Glycine max using BLASTp. The *GmUFO1* gene fragment was amplified from soybean cultivar Williams 82, and the primers were designed using Primer5 software (https://primer.software.informer.com/5.2/, accessed on 3 January 2021) (Appendix A). PCR amplifications were performed using 2 × Phanta Max Master Mix (Dye Plus) (Vazyme Biotech Co., Nanjing, Jiangsu, China) DNA polymerase in a 50 μL mix containing 2 μmol L^−1^ of each primer, 25 units of 2 × Phanta Max Master Mix (Dye Plus), and 1 μL of cDNA. The PCR experiments were performed as follows: 95 °C for 30 s, 32 cycles of 95 °C for 15 s, 56 °C for 15 s, 72 °C for 2 min, and 72 °C for 5 min. The PCR products were cloned using the TA cloning vector pGEM-Teasy (Promega Biotech Co., Beijing, China) and sequenced. After verifying the sequences, these products were used to construct the in situ hybridization and subcellular localization vector.

### 4.3. Phylogenetic Analysis and Physical Properties of UFO Genes

Phylogenetic analysis was performed using MEGA11 software (Version: 7.0, MEGA software development team, Tempe, AZ, USA) [31] and following the neighbor-joining method to analyze the evolutionary relationships among the 11 *UFO* genes from nine plants, including both monocots and dicots, namely *Arabidopsis*, soybean, *Medicago*, cowpea, kidney bean, cotton, rice, maize, and barley.

ProtParam (https://web.expasy.org/protparam/database, accessed on 10 March 2021) was used to determine the physical properties of the UFO proteins, such as the molecular weight, the length of each protein, and the protein isoelectric point (PI). 

### 4.4. In Situ Hybridization

The tissue fixation for in situ hybridization was performed following the protocol of Feng et al. [32]. The tissues were embedded in paraffin (Paraplast Plus, Sigma-Aldrich, Saint Louis, MO, USA) and sliced into 8 μm sections with a microtome (Leica, Wetzlar, Germany). The 3′-region of *GmUFO1* cDNA was subcloned and used as a template to generate sense and antisense RNA probes (Appendix A). Digoxigenin-labeled RNA probes were prepared with a DIG RNA Labeling Kit (T7/SP6) (Cat. No. 11175025910, Roche, Basel, Switzerland) according to the manufacturer’s instructions. The slides were observed under a bright field using a Leica microscope (DMI8, Leica, Wetzlar, Germany).

### 4.5. Subcellular Localization

For the GmUFO protein subcellular experiment, the full-length CDS (removal terminator) sequence of *GmUFO1* was amplified and linked with a GFP fluorescent protein sequence and inserted into the *pUC19* vector using NdeΙ and SalΙ for double digestion. The subcellular expression vector *pUC19-GFP-GmUFO1* was constructed containing the 35S promotor controlling expression of the *GFP-GmUFO1* fusion construct. A modified polyethylene glycol (PEG) method was used for the transfection of protoplasts [33,34,35]. The *pUC19-GFP-GmUFO1* and *pUC19-AHL22-mCherry* plasmids were extracted by a TIANprep Mini Plasmid Kit (TIANGEN Biotech Co., Beijing, China), and the two plasmids were transiently transformed into *Arabidopsis* protoplasts. These transformed protoplasts were subsequently cultured at room temperature in the dark for 16 h, and fluorescence observation and photography were performed using a Leica microscope (DMI8, Leica, Wetzlar, Germany).

### 4.6. Cytological Analysis

The floral primordium of wild-type Williams 82 during five developmental stages (stages 1–5) was observed by scanning electron microscopy. After removing the bracts, they were fixed in a formaldehyde alcohol acetic acid (FAA) fixation solution for at least 12 h. All materials were dehydrated with ethanol (30, 50, 70, 90, 95, 100%) and then dehydrated with ethanol-tert-butanol to 100% tert-butanol. The samples were dissected using a positive microscope and held in small Petri dishes with conductive tape. The floral primordium was scanned using InTouchScope Scanning Electron (JSM-IT500, Japan Electronics, Tokyo, Japan). The anatomical observation of flower organs was performed using an orthomicroscope (SZ2-ILST, Olympus Co., Tokyo, Japan) which was photographed with Olympus cellSens Standard, imaging software (Version 1.16, Olympus Co., Tokyo, Japan).

### 4.7. Gene Knockout and Material Identification

A Cas9/sgRNA plasmid construction kit (Viewsolid Biotech Co., Beijing, China) was used to construct the CRISPR/Cas9 plasmids. These plasmids contained a dicotyledon codon-optimized dpCas9 (under the *GmUbi3* promoter) and a gRNA scaffold (under the *GmU6-2* promoter). The construction of the CRISPR/Cas9 vector was performed following the manufacturer’s instructions (Viewsolid Biotech Co., Beijing, China). The constructed CRISPR/Cas9-gDNA plasmid was transformed into Agrobacterium EHA105, and soybean genetic transformation was performed following Yamada et al. [36]. Genomic DNA was extracted from the seedlings of wild-type and all positive transgenic plants, and DNA fragments spanning the target sites were amplified by PCR. The target site for *GmUFO1* was amplified with the primers GmUFO-F (OL13180) and GmUFO1-R (OL13181). The target site for *GmUFO2* was amplified with the primers GmUFO-F (OL13180) and GmUFO2-R (OL14387). The PCR products were sequenced using primer OL13180. After sequencing, four T0 generation mutant lines (SP2464, SP2465, SP2466, and SP2467) were identified, and then the seeds were sowed to generate T1 generation plants, which were all sequenced to identify single and double mutants. The pedigree of the T0 and T1 generation are listed in Appendix A, and the sequencing results of T0 and T1 generation mutants are listed in Appendix A. The primers are listed in Appendix A.

### 4.8. Total RNA Extraction and qPCR

Total RNA was extracted using the Trizol (TIANGEN Biotech Co., Beijing, China) method [37], and the cDNA was synthesized by the process of reverse transcription using the TransScript One-Step genomic DNA (gDNA) removal and cDNA synthesis Super-Mix kit (TransGen Biotech Co., Beijing, China), with gDNA removal in one step. The cDNA was diluted in sterile water to 100 ng/μL, with a 2 μL diluent as a template for qPCR. qPCR was performed using 2 × RealStar Green Fast mix (GenStar Biosolutions Co., Beijing, China) on a Stratagene Mx3005P sequence detection system (Stratagene, La Jolla, CA, USA) following the manufacturer’s instructions. Three biological replicates were used to quantify the expression levels. The differences between the groups were calculated using the 2^−ΔΔCt^ method. The soybean *GmActin 11* (*Glyma.18G290800*) gene was used as an internal control, and the relative expression levels were calculated. The primers used for RT-qPCR analysis are shown in Appendix A. 

## Figures and Tables

**Figure 1 ijms-24-09662-f001:**
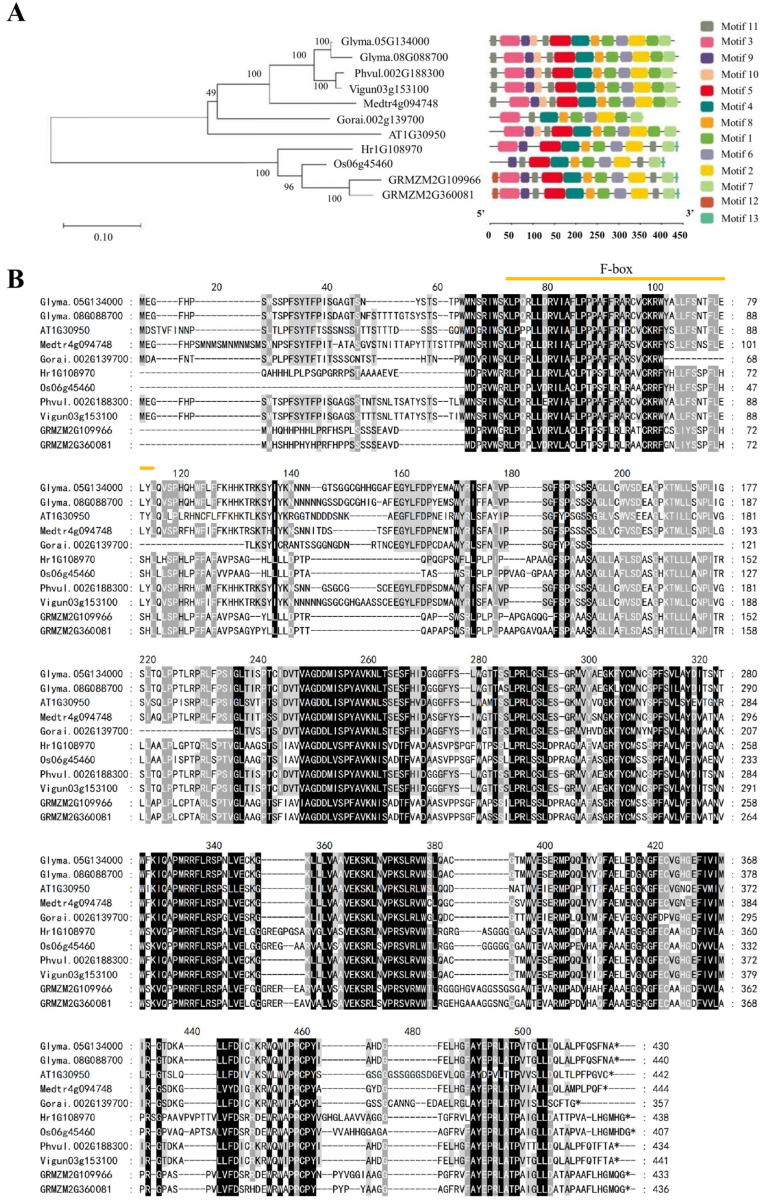
Phylogenetic tree and multiple sequence alignment of *UFO* genes. (**A**) Phylogenetic analysis of the selected 11 UFO proteins from nine species, namely *Arabidopsis thaliana* (AT1G30950), *Glycine max* (Glyma.05G134000, Glyma.08G088700), *Medicago truncatula* (Medtr4g094748), *Vigna unguiculate* (Vigun03g153100), *Phaseolus vulgaris* (Phvul.002G188300), *Gossypium raimondii* (Gorai.002G139700), *Oryza sativa* (Os06g45460), *Zea mays* (GRMZM2G109966, GRMZM2G360081), and *Hordeum vulgare* (Hr1G108970). The neighbor-joining tree was constructed using MEGA11, and the bootstrap value was set at 1000 replicates. (**B**) Amino acid sequence alignment of UFOs in soybean and other species. The orange line represents the F-box domain; the black shade represents the conserved amino acids, and the gray shade represents the relatively conserved amino acids.

**Figure 2 ijms-24-09662-f002:**
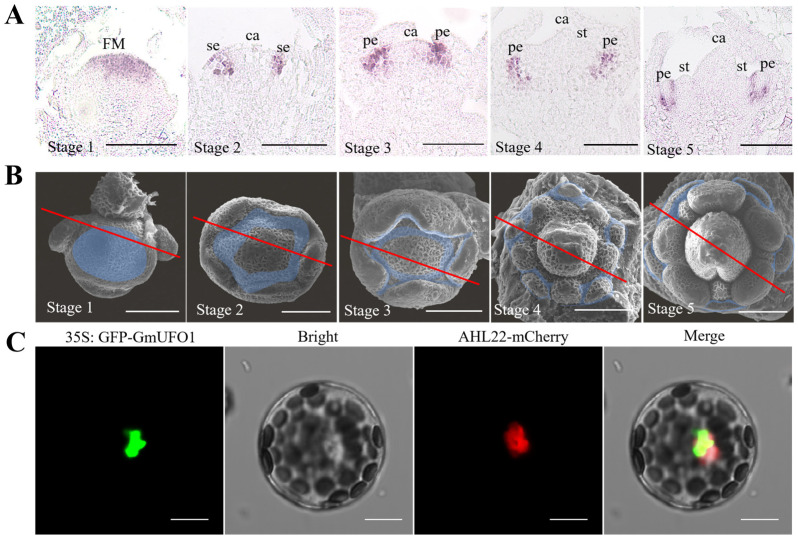
The expression profile of *GmUFO1*. (**A**) In situ hybridization of the *GmUFO1* gene in the soybean flower primordium of different development stages, and the signals were purple. Bar = 100 μm. (**B**) Scanning electron microscopy (SEM) observations of in situ hybridization and stereoscopic models of *GmUFO1* gene expression at five developmental stages. The areas of gene expression were shown in blue and the red lines represented the slice position corresponding to (**A**). Bar = 100 μm. (**C**) Expression location of GmUFO1-GFP in *Arabidopsis thaliana*. The GFP signals were green and mCherry signals were red. Bar = 10 μm.

**Figure 3 ijms-24-09662-f003:**
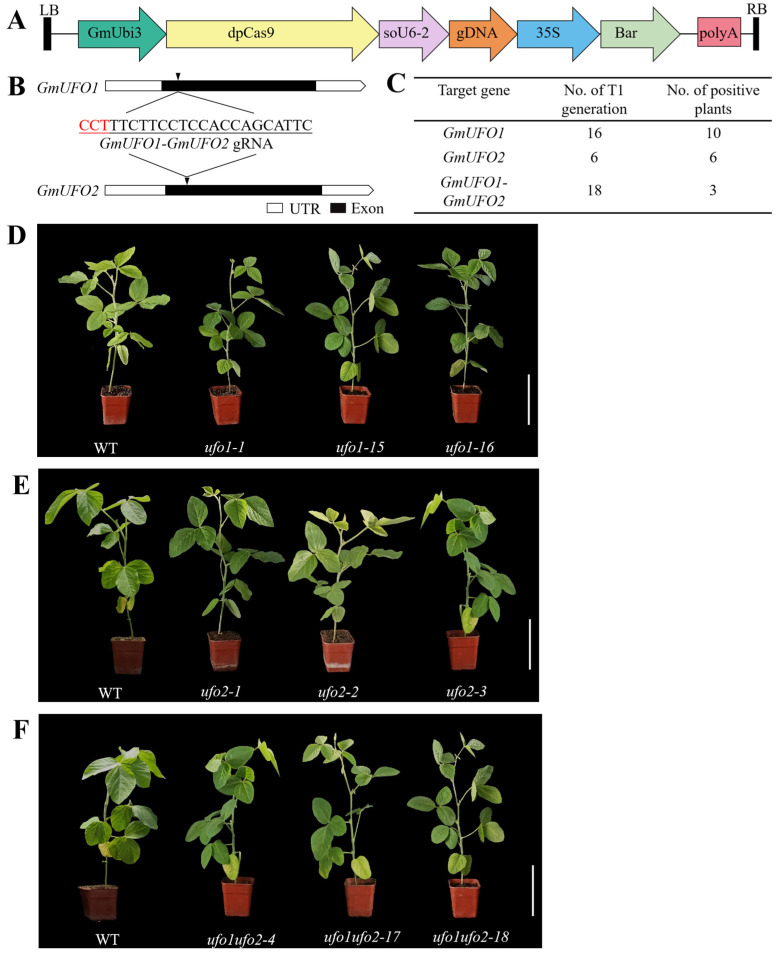
Summary of CRISPR/Cas9-mediated genome editing of *GmUFO* genes. (**A**) Structural diagram of the CRISPR/Cas9 vector. (**B**) Target sites designed for *GmUFO1* and *GmUFO2* in CRISPR/Cas9, and the PAM sequence was in red. (**C**) Summary of mutation frequency in the T1 generation. (**D**–**F**) Seedling phenotype of *Gmufo1* single mutant lines (**D**), *Gmufo2* single mutant lines (**E**), and *Gmufo1ufo2* double mutant lines (**F**) compared with the wild type. Scale bar = 10 cm.

**Figure 4 ijms-24-09662-f004:**
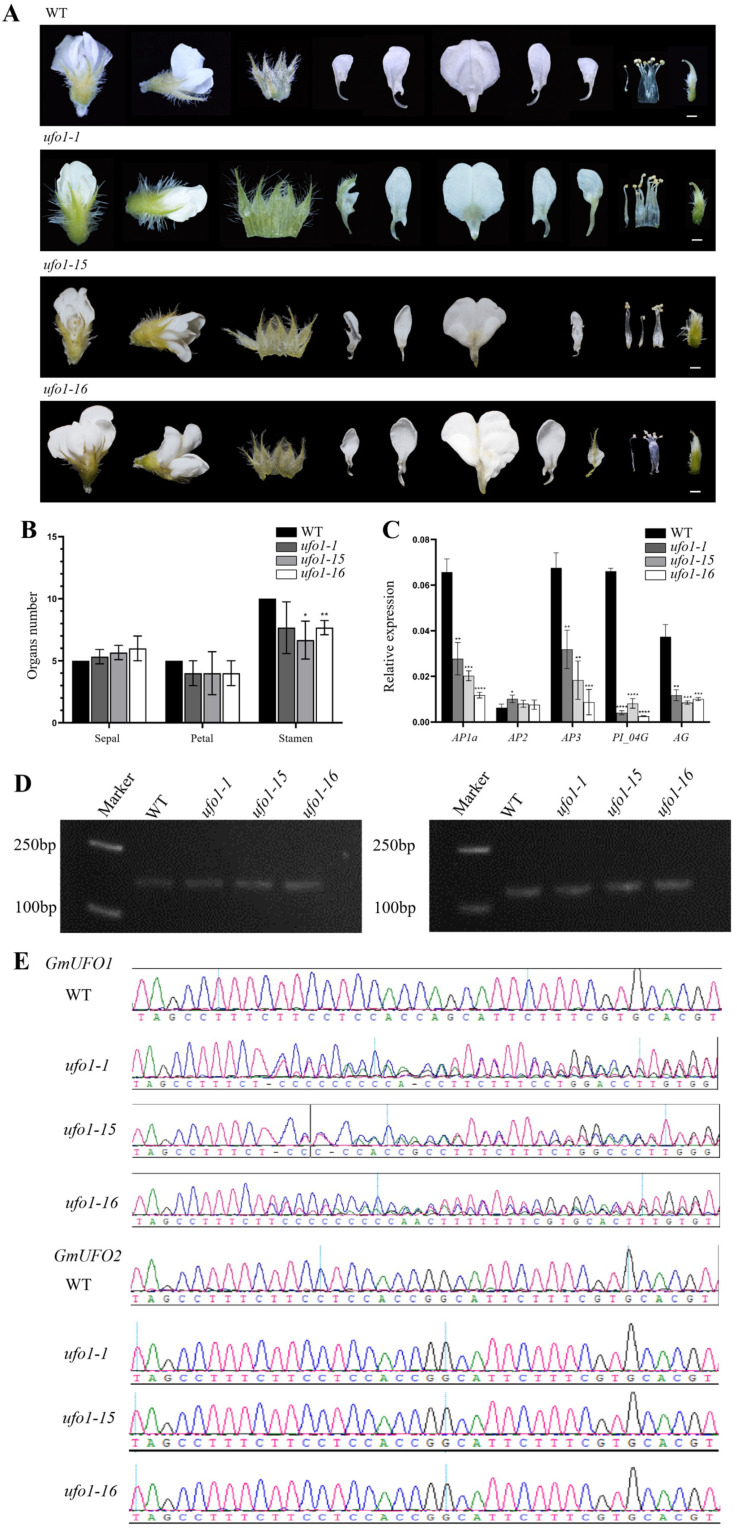
Identification of the *GmUFO1* knockout lines. (**A**) Floral organ phenotype of the three *Gmufo1* mutant lines compared with the wild type. Scale bar = 1 mm. (**B**) Statistics of the number of sepals, stamens, and petals in *Gmufo1* mutant lines. Significant differences according to two-sided Student’s *t* test (* *p* < 0.05, ** *p*  <  0.01). Data are means ±  SD for at least 10 flowers in each line. (**C**) ABC function gene expression was analyzed in the flowers of Williams 82, *Gmufo1-1*, *Gmufo1-15*, and *Gmufo1-16*. All data presented are mean ± SD for three biological replicates. Asterisks indicate significant differences according to two-sided Student’s *t* test (* *p* < 0.05, ** *p* < 0.01, *** *p* < 0.001, **** *p* < 0.0001). (**D**) *GmUFO1* and *GmUFO2* expression level in three mutant lines and the wild type. (**E**) CRISPR/Cas9 target site sequencing of *GmUFO1* and *GmUFO2* in the three selected lines.

**Figure 5 ijms-24-09662-f005:**
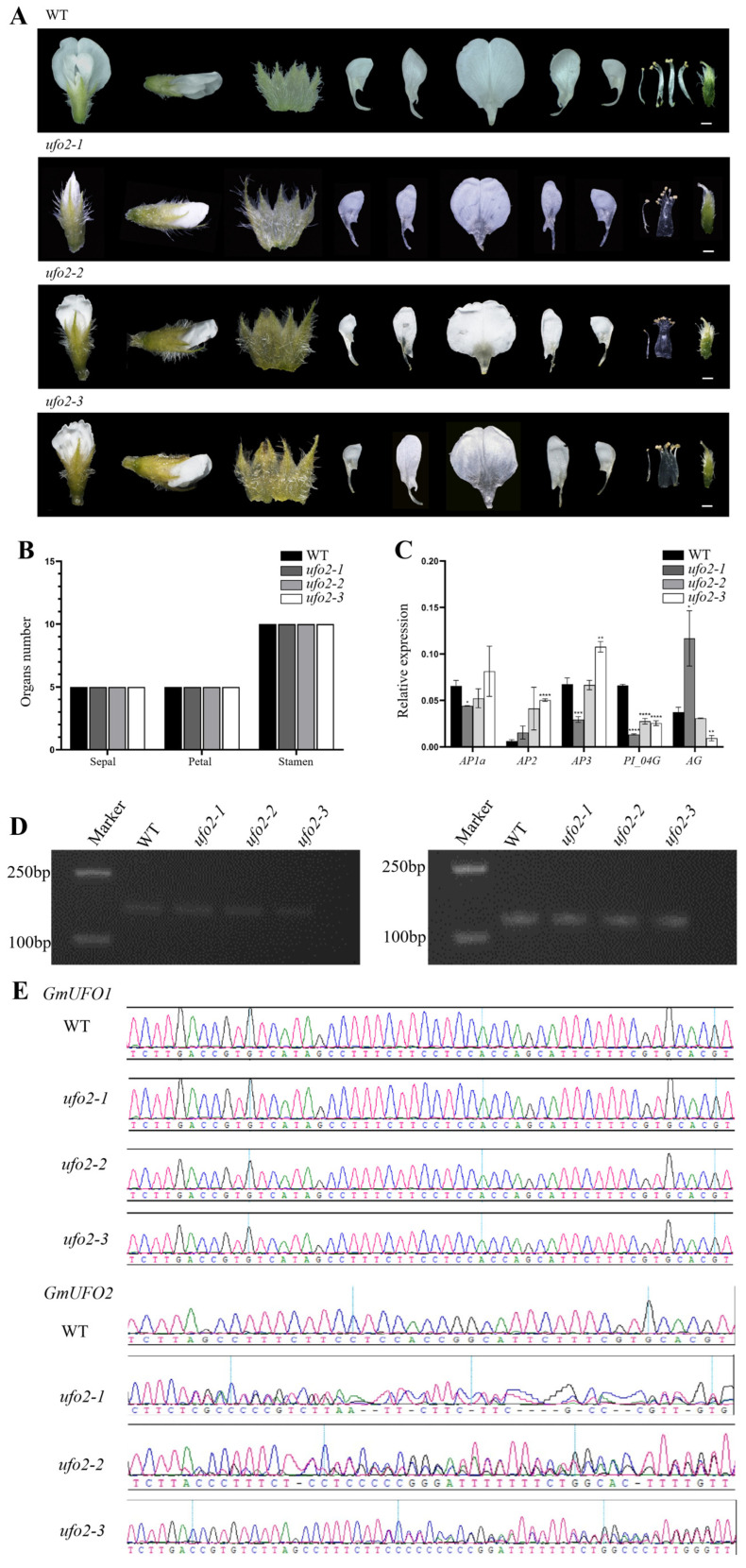
Identification of the *GmUFO2* knockout lines. (**A**) Floral organ phenotype of the three *Gmufo2* mutant lines compared with the wild type. Scale bar = 1 mm. (**B**) Statistics of the number of sepals, stamens, and petals in *Gmufo2* mutant lines. Non-significant differences were found according to two-sided Student’s *t* test. Data are means ±  SD for at least 10 flowers of each line. (**C**) ABC model gene expression was analyzed in the flowers of Williams 82, *Gmufo2-1*, *Gmufo2-2*, and *Gmufo2-3*. All data presented are mean ± SD for three biological replicates. Asterisks indicate significant differences according to two-sided Student’s *t* test (* *p* < 0.05, ** *p* < 0.01, *** *p* < 0.001, **** *p* < 0.0001). (**D**) *GmUFO1* and *GmUFO2* expression level in three mutant lines and the wild type. (**E**) CRISPR/Cas9 target site sequencing of *GmUFO1* and *GmUFO2* in the three selected lines.

**Figure 6 ijms-24-09662-f006:**
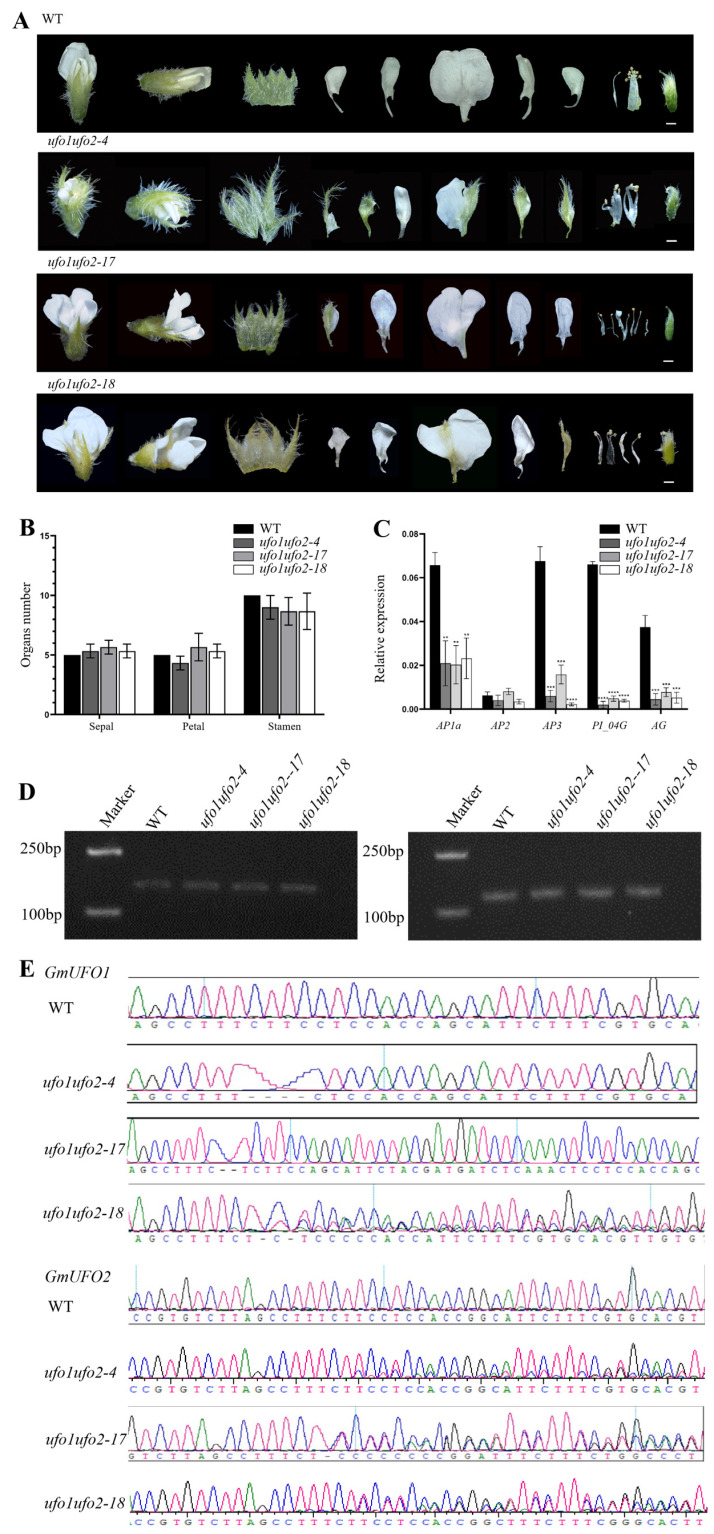
Identification of the *GmUFO1* and *GmUFO2* knockout lines. (**A**) Floral organ phenotype of the three *Gmufo1ufo2* mutant lines compared with the wild type. Scale bar = 1 mm. (**B**) Statistics of the number of sepals, stamens, and petals in the *Gmufo1ufo2* mutant lines. Non-significant differences were found according to two-sided Student’s *t* test. Data are means ±  SD for at least 10 flowers in each line. (**C**) ABC model gene expression was analyzed in the flowers of Williams 82, *Gmufo1ufo2-4*, *Gmufo1ufo2-17*, and *Gmufo1ufo2-18*. All data presented are the means ± SD for three biological replicates. Asterisks indicate significant differences according to two-sided Student’s *t* test (** *p* < 0.01, *** *p* < 0.001, **** *p* < 0.0001). (**D**) *GmUFO1* and *GmUFO2* expression level in three mutant lines and the wild type. (**E**) CRISPR/Cas9 target site sequencing of *GmUFO1* and *GmUFO2* in the three selected lines.

## Data Availability

The datasets used in the study are available from the corresponding author upon request.

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
