# Peer review of "GmUFO1 Regulates Floral Organ Number and Shape in Soybean"

_ijms, 2023, doi:10.3390/ijms24119662_

Round 1

Reviewer 1 Report

The manuscript (MS) showed soybean UFO as other plant UFO regulates floral organ development, especially in the second whorl organ. Using CRISPR/Cas9 the author investigated function of each UFO on flower development in soybean. Although it is not a novel study, the MS indicates that the function of UFO is conserve among plants, particularly in dicot plants. However, there are some major concerns that the authors should clarify.

In this study, the authors mainly focused on GmUFO1, however they did not mention why they chose GmUFO1 for Spatial and temporal expression profiles in the MS. Thus, I strongly recommend the authors should carry the In situ hybridization of the GmUFO2 gene in the soybean flower primordium as they did for GmUFO1.

For CRISPR experiments, the authors obtained four T0 independent transgenic lines, however, they did not genotype these T0 mutants. Therefore, it is unclear whether the T1 plants of gmufo1, gmuof2 and the double mutant gmufo1/ufo2 are generated from the same or different transgenic lines. Addition, without genotyping for the T0 plants, it is unclear how do the authors can generate each mutant lines. Therefore, the authors should clearly describe the method, primers to generate the T1 mutants in the method and result sections of the MS.

Knock-out ufo showed abnormal in floral organ number in soybean. Therefore, the author should carry out the yield test of the mutant lines as compare to the wild-type.

The authors should do statistical analysis for Figure 4 BC, Figure 5 B,C and Figure 6 B,C.

Author Response

Dear Reviewer,

Thank you very much for your comments, and it helps to improve the manuscript greatly. We have carefully supplemented the data according to your comments. The reply of each comment and the results were listed in the file, Please see the attachment. 

Sencerely yours,

Yaohua Zhang

Reviewer 2 Report

GmUFO1 regulates floral organ number and shape in soybean

Identification and clarification of the molecular mechanism of the genes involved in the regulation of the reproductive period and flower development in soybean makes this topic original and relevant, having in mind that reproductive stage in soybean plays a key role in seed yield formation and achieving a good grain quality. The main contribution of this work in solving the mentioned problem is in identification of UFO genes in soybean, where   GmUFO1 plays a key role in the regulation of flower organ formation, while GmUFO2 have possible indirect role in floral development, through the interaction with GmUFO1.

The methodology of performing experiments is appropriate and very well, concisely and clearly written in the manuscript. Every step, starting with identification of putative orthologs of UFO genes in soybean, designing primers, RNA extraction and qPCR,  phylogenetic analysis of UFO genes in  Arabidopsis, soybean, medicago, cowpea, kidney bean, cotton, rice, maize and barley, hybridization, cytological analysis using an electron microscope, genetic transformation and gene knock-out were performed correctly. Discussion is extensive, experienced and supported by the up to date references. Conclusions are consistent with the evidence and the results presented. The results are presented clearly, the only suggestion I would give to authors to try to shorten a text in figure titles. Pictures are clear and prepared in good resolution.

L 197 "We also detected..." suggestion: try to avoid "we" in scientific manuscript. "Also" is a redundant word, by my opinion

Author Response

Dear professor,

Thank you very much for your comments, and it helps to improve the manuscript greatly. The reply for your comments are as follows:

Comment 1: try to shorten a text in figure titles.

Reply: thanks for your comment, and we have shortened the text in figure titles.

Comment 2: L 197 "We also detected..." suggestion: try to avoid "we" in scientific manuscript. "Also" is a redundant word, by my opinion

Reply: Thanks very much for your advice, and we have altered all of the “We…” sentences in the manuscript.

Sencerely yours,

Yaohua Zhang

Round 2

Reviewer 1 Report

After revised, the MS should be accepted for publication.